# Enzymatic Characterization of Wild-Type and Mutant Janus Kinase 1

**DOI:** 10.3390/cancers11111701

**Published:** 2019-11-01

**Authors:** Nicholas P. D. Liau, Artem Laktyushin, Rhiannon Morris, Jarrod J. Sandow, Nicos A. Nicola, Nadia J. Kershaw, Jeffrey J. Babon

**Affiliations:** 1Walter and Eliza Hall Institute, 1G Royal Parade, Parkville 3052, VIC, Australia; laktyushin@wehi.edu.au (A.L.); morris.r@wehi.edu.au (R.M.); sandow@wehi.edu.au (J.J.S.); nicola@wehi.edu.au (N.A.N.); kershaw@wehi.edu.au (N.J.K.); 2Department of Medical Biology, The University of Melbourne, Royal Parade, Parkville 3050, VIC, Australia

**Keywords:** jak, jak/stat, myeloproliferative disease, v617f, jak, jak/stat, myeloproliferative disease, v617f

## Abstract

Janus kinases (JAKs) are found constitutively associated with cytokine receptors and are present in an inactive state prior to cytokine exposure. Activating mutations of JAKs are causative for a number of leukemias, lymphomas, and myeloproliferative diseases. In particular, the JAK2^V617F^ mutant is found in most human cases of polycythemia vera, a disease characterized by over-production of erythrocytes. The V617F mutation is found in the pseudokinase domain of JAK2 and it leads to cytokine-independent activation of the kinase, as does the orthologous mutation in other JAK-family members. The mechanism whereby this mutation hyperactivates these kinases is not well understood, primarily due to the fact that the full-length JAK proteins are difficult to produce for structural and kinetic studies. Here we have overcome this limitation to perform a series of enzymatic analyses on full-length JAK1 and its constitutively active mutant form (JAK1^V658F^). Consistent with previous studies, we show that the presence of the pseudokinase domain leads to a dramatic decrease in enzymatic activity with no further decrease from the presence of the FERM or SH2 domains. However, we find that the mutant kinase, in vitro, is indistinguishable from the wild-type enzyme in every measurable parameter tested: K_M_ (ATP), K_M_ (substrate), k_cat_, receptor binding, thermal stability, activation rate, dephosphorylation rate, and inhibitor affinity. These results show that the V658F mutation does not enhance the intrinsic enzymatic activity of JAK. Rather this data is more consistent with a model in which there are cellular processes and interactions that prevent JAK from being activated in the absence of cytokine and it is these constraints that are affected by disease-causing mutations.

## 1. Introduction

Cytokines that signal through the JAK/STAT pathway are important modulators of hematopoiesis, inflammation, and the immune response [1,2]. Cytokines are small secreted proteins that bind to the extracellular domains of specific receptors on the surface of target cells to initiate a cytokine-specific transcriptional program. Cytokine receptors lack intrinsic kinase domains and so rely on the constitutively associated Janus kinases (JAKs) to phosphorylate downstream substrates and initiate the signaling cascade [3]. JAKs phosphorylate tyrosine residues on the cytoplasmic domains of the receptors to which they are attached, and these newly phosphorylated sites recruit downstream effector proteins and transcription factors, the most important of which are the signal transducers and activators of transcription (STATs). STATs are in turn activated by JAK-mediated phosphorylation and then dissociate from the receptor and migrate into the nucleus to initiate the cytokine-induced transcriptional program [4].

There are four JAK family members—JAK1, JAK2, JAK3, and TYK2, which all share the same basic domain architecture (see Figure 1A), in addition to 23% overall sequence identity and 48% overall similarity [5]. JAK2 is known to be important for hematopoiesis, as JAK2 knockout mice die before birth as a lack of erythrocytes [6,7]. JAK1, JAK3, and TYK2 are known to be important for development of the immune system, though a JAK1 knockout results in a more severe phenotype than JAK3 or TYK2, suggesting that JAKs are at least partially redundant for some functions [8,9,10,11]. The N-terminal half of the protein contains a FERM-SH2 module which anchors the protein to the intracellular Box1 and Box2 motifs of a cytokine receptor [12,13]. The C-terminal half consists of two kinase domains, the first of which is catalytically inactive (termed the JH2 or pseudokinase domain) whilst the second is catalytically active (the JH1 or kinase domain) and phosphorylates downstream substrates.

JAKs are found to be associated with receptors but in an inactive state prior to cytokine exposure. Their kinase domains contain a structural feature known as the activation loop, which contains two consecutive tyrosine residues. When the first of these tyrosines is unphosphorylated the kinase is inactive. Phosphorylation of this tyrosine leads to re-orientation of the activation loop and activates the domain. Phosphorylation of this tyrosine is catalyzed in trans by the kinase domain of the neighboring JAK molecule attached to the other chain of the cytokine receptor [3]. This auto-activation occurs in response to cytokine engaging the extracellular domain of the receptor and inducing either dimerization (or multimerization) [14,15] or reorientation [16]. The pseudokinase domain plays an important part in regulating the activation process since its removal results in constitutive partial activation of the kinase as well as a failure to fully-activate in the presence of cytokine [17]. The importance of this domain is highlighted by the fact that the majority of activating (and therefore disease-causing) mutations of JAKs are found in this domain.

The most common disease-causing JAK mutation is the V617F point mutation in the pseudokinase domain of JAK2. This mutation is causative for a group of diseases known as the myeloproliferative neoplasms (MPNs), [18,19,20,21], particularly the disease Polycythemia Vera in which there is an overproduction of erythrocytes. Other, less common, JAK2 MPN mutations are clustered in exon 12, corresponding to the region linking the pseudokinase domain and the SH2 domain [22,23], whereas mutations in exon 16, which correspond to residues in the pseudokinase domain, are associated with acute lymphoblastic leukemias [24,25,26]. The common molecular phenotype induced by these mutations is that JAK is activated even in the absence of a cytokine and can, therefore, promote cytokine-independent signaling. In cells, however, this appears to require the presence of JAK-binding homodimeric cytokine receptors [27,28]. The equivalent of the JAK2 V617F mutation in JAK1 (JAK1 V658F) has been linked with other cancers [29,30]. This mutation has also been shown to result in constitutive JAK activation in cell lines [31].

The molecular mechanism by which JAK is kept in an inactive state prior to cytokine exposure and thence activated is not fully understood, neither is it understood how mutations lead to constitutive activation. Although structural details of the FERM-SH2 and pseudokinase/kinase domains are available [12,32,33] there is no structure of the full-length protein. Detailed kinetic studies have not been performed on the full-length protein, either wild-type or mutant due to the difficulty of producing it recombinantly. We have developed methods for the expression and purification of recombinant full-length JAK1 WT and the JAK1 V658F oncogenic mutant, allowing enzymatic analysis of the wild-type and mutant forms of the enzyme. Here, we present an enzymatic characterization of these two forms of the protein in addition to constructs lacking the upstream FERM/SH2 and pseudokinase regulatory domains. We show that the presence of the pseudokinase domain leads to a dramatic decrease in enzymatic activity but that this difference is not affected by the presence of the V658F mutation. In fact, we show that the V658F point mutant is indistinguishable from the wild-type enzyme in any parameter we could measure: K_M_ (ATP), K_M_ (substrate), k_cat_, receptor binding, thermal stability, activation rate, dephosphorylation rate, or inhibitor affinity. Our data show that the V658F mutation does not affect intrinsic enzymatic parameters. Thus, its aberrant activation in cells is likely due to a failure in the interactions, such as those with the cytokine receptor to which it is bound, that prevent it from being activated (in the absence of cytokine) in the first place.

## 2. Results

### 2.1. Expression and Purification of JAK1

After an extensive screening of different constructs, expression conditions, and purification protocols, we developed a protocol capable of producing JAK1 at the quantities and purity required for enzymatic analysis. All constructs were produced by expression in *Sf*21 insect cells following baculovirus infection. The key optimization factors for the full-length enzyme (which was the most difficult to produce) were the deletion of the 25 N-terminal residues, infection of *Sf*21 cells with a high MOI (>3), and performing all purification steps in buffers with the addition of 10% (*v*/*v*) glycerol, 500mM NaCl and 2mM TCEP. This procedure was used to express both wild-type and mutant full-length (ΔN25) constructs in addition to FERM/SH2, pseudokinase/kinase and kinase domains. The final products were ~90% pure as judged by SDS-PAGE analysis (Figure 1B). Similar attempts to produce full-length, recombinant, JAK2 were not successful.

### 2.2. The Pseudokinase Domain Decreases the Turnover Rate of JAK1

In order to determine the catalytic rate, k_cat_, of the purified wild-type and mutant JAK constructs we first needed to (A) phosphorylate the kinase to allow it to become activated and (B) then quantify the amount of active enzyme in each of the preparations. In vivo, JAKs autophosphorylate each other in trans via a receptor-mediated reaction. However, this can be mimicked in vitro by prolonged incubation with ATP and Mg^2+^ to ensure activation loop phosphorylation. After this was performed, the determination of the amount of active enzyme was done by active-site titration [34] with the tight-binding small-molecule JAK1 inhibitor itacitinib [35,36]. The concentration of active enzyme was then calculated by fitting the resulting enzymatic activities using the Morrison equation [34] (Figure 2A):$$\frac{V_{i}}{V_{0}}=1-\frac{\left(\left[E\right]+\left[I\right]+K_{i}^{app}\right)-\sqrt{{\left(\left[E\right]+\left[I\right]+K_{i}^{app}\right)}^{2}+4\left[E\right]\left[I\right]}}{2\left[E\right]}$$
where
$$K_{i}^{app}=K_{i}\left(1+\frac{\left[S\right]}{K_{M}}\right)$$
and [E] is the active JAK1 concentration, [I] is itacitinib concentration, [S] is ATP concentration, and K_M_ refers to K_M_ (ATP). Vi and Vo refer to enzyme velocity in the presence and absence of inhibitor, respectively.

Once the active-site titration was performed, each JAK construct was then used in a kinase assay to determine the catalytic rate. High concentrations of ATP and STAT peptide substrate (1 mM and 5 mM respectively) were used and enzymatic activity was calculated and normalized to the concentration of active enzyme present to yield k_cat._ As shown in Figure 2B, full-length JAK1 has a turnover rate of 0.014 s^−1^ which is approximately 30-fold slower than the kinase domain alone (k_cat_ 0.4 s^−1^). The presence of the pseudokinase domain alone was responsible for this reduction in turnover rate as the pseudokinase/kinase constructs had an identical k_cat_ to the full-length enzyme, showing that the FERM/SH2 domains do not influence enzymatic activity. The reduction in catalytic rate we observed for the pseudokinase/kinase domain is consistent with previous studies [17,37,38]. However, there were no differences in k_cat_ between any of the wild-type and mutant forms of the enzyme, indicating that the V658F mutation does not lead to an increase in intrinsic enzymatic activity.

### 2.3. The V658F Mutation Has No Effect on ATP or Substrate Affinity

In order to determine the effect of the V658F mutation on substrate binding we performed steady-state kinetics and measured the Michaelis–Menten constant (K_M_) for both ATP and peptide substrate (STAT5b peptide) for each construct. K_M_ indicates the concentration of a particular substrate at which an enzyme reaches half its maximal velocity and is a rough surrogate for substrate affinity.

To measure K_M_ (peptide) a concentration of 0.5 mM ATP was used, later shown to be 10-fold higher than K_M_(ATP). Results showed that K_M_ (peptide) was approximately 5mM for the full-length enzyme and that there was no significant difference in this value for any truncated constructs of the enzyme (Figure 3A,B). Likewise, there was no change in K_M_ (peptide) when the mutant form of the enzyme was tested. This indicates that the affinity for the peptide substrate is not affected by the pseudokinase domain, FERM-SH2 domains, or the V658F mutations.

As K_M_ (peptide) is high, we were unable to use saturating amounts of this substrate (due to solubility limits) to measure K_M_ (ATP). Therefore, we instead measured K_M, apparent_ (ATP) in kinase assays where the concentration of the peptide substrate was 6 mM. As shown in Figure 3C,D, K_M, app_ (ATP) for the full-length enzyme was 25 µM, which was an order of magnitude lower than that measured for the kinase domain alone (175 µM). Once again, the presence of the V658F mutation had no effect on K_M, app_ (ATP) and neither did the FERM/SH2 domain affect this parameter.

### 2.4. The V658F Mutation Has No Effect on Autophosphorylation or Dephosphorylation Rates In Vitro

A known effect of the JAK2 V617F mutation and its orthologous mutation in other members of the JAK family (such as JAK1 V658F) in vivo is to cause constitutive activation of the kinase even in the absence of cytokine stimulation [27,28,39]. To determine whether the V658F mutation causes an enhancement in the activation rate, we measured the activation of the wild-type and mutant enzymes over time. We purified batches of JAK1 FL WT and JAK1 FL V658F in the absence of ATP/Mg^2+^. This resulted in a sample that was largely unphosphorylated as determined by western blot, although some phosphorylation had occurred in the expression host (Figure 4A). Notably, there was no difference in basal levels of phosphorylation between the wild-type and mutant enzyme within the expression host. The purified enzyme at concentrations of 5 µM and 15 µM were then incubated with ATP and MgCl_2_ and activation loop phosphorylation measured by western blot using a phospho-specific antibody. JAK1 activation loop phosphorylation over time was assayed by western blot. As expected, JAK activation reached saturation more quickly at higher protein concentrations (Figure 4A), indicating that substrate availability (JAK itself in this case) is the rate-limiting step to autoactivation in solution; however, there was no reproducible difference between the activation rates of the mutant and wild-type enzymes when measured in vitro.

Another possibility for the increased activity of the V658F mutation observed in vivo might be increased resistance to phosphatases. To determine whether the V658F mutation protects against dephosphorylation we incubated the enzyme with the phosphatase PTP1b and measured dephosphorylation over time. As shown in Figure 4B there was no difference in susceptibility to phosphatase action in vitro between the wild-type and mutant forms of the enzyme.

### 2.5. The V658F Mutation Does Not Affect the Thermal Stability of JAK1

To determine whether the V658F mutation affects protein stability we performed thermal shift assays on all the JAK constructs in the presence and absence of various cofactors that bind the kinase domain. SYPRO orange, which fluoresces when bound to hydrophobic residues, was added to a solution of protein and ligand. Fluorescence was measured as the temperature of the solution was increased, with increased fluorescence indicating the unfolding of the protein. The melting temperature (T_m_) is represented by the inflection point of the fluorescence curve, with a higher T_m_ indicating greater thermal stability [40].

The T_m_ of JAK1 constructs was measured in the presence of EDTA, which is expected to disrupt ATP binding by chelation of Mg^2+^. Full-length JAK1 and the pseudokinase-kinase domain exhibited a slightly higher T_m_ than the kinase domain (Figure 5). However, there was no significant difference in T_m_ between any of the JAK1 V658F mutants when compared to their wild type equivalent. In the presence of ATP/Mg^2+^, all constructs containing a kinase domain exhibited an increased T_m_ compared to EDTA, with a further T_m_ increase observed upon addition of Itacitinib (an inhibitor which binds the ATP-binding site). However, there was no difference in the thermal stability of the wild-type and mutant forms of JAK1 in the presence or absence of any cofactors.

### 2.6. Cytokine Receptor Box1/Box2 Binding Affinity Is Not Affected by Either the Pseudokinase and Kinase Domains or the V658F Mutation

We considered the possibility that the pseudokinase and kinase domains of JAK may affect the binding of JAK to the cytokine receptor (either directly or allosterically), as well as the possibility that the V658F mutation might exert its oncogenic effect by influencing the interaction of JAK with receptor. It has been previously reported that the JAK1 FERM-SH2 domain binds with high affinity (K_d_ = 71 nM) to the interferon-lambda receptor (IFNλR) [41]. We performed isothermal titration calorimetry (ITC) to determine the affinity of a peptide representing the IFNλR Box1 and Box2 motifs (human residues K250 - T299). As shown in Figure 6, we found the JAK1 FERM-SH2 domain binds this motif with near-identical affinity to that previously reported and that this value was not significantly different when full-length JAK1 was tested. However, there was no significant effect on receptor binding between the wild-type and mutant forms of the enzyme.

Receptor-binding is an important regulator of the activity of JAK in vivo and may be required in order for the V658F mutation to induce constitutive activity [27,28]. Although we cannot include full-length receptor constructs in these assays we were able to include a peptide fragment that represents the box1/2 motifs. Therefore, we performed steady-state kinetics using JAK1 wild-type and V658F in the presence and absence of saturating concentrations of the IFNλR peptide. For both constructs, no significant differences in ATP K_M_ or turnover rate were measured with or without receptor peptide (Figure 7). This shows that receptor peptide-binding does not affect these kinetic parameters of JAK1 and that occupation of the FERM/SH2 domain receptor binding site does not lead to a change in the enzymatic activity of the V658F mutant.

### 2.7. Wild-Type and Mutant JAK1 Are Equally Susceptible to Itacitinib Inhibition

To our knowledge, it is unknown whether small-molecule JAK inhibitors bind the wild-type and mutant forms of the enzyme with equal affinity. Therefore, the inhibition constant (K_i_) for Itacitinib was determined. JAK1 activity was measured by kinase assay in the presence of increasing concentrations of Itacitinib. As shown in Figure 8 the V658F mutation did not affect inhibitor binding and all constructs were inhibited by itacitinib with K_i_ between 0.5 nM and 2 nM. Although there was a tendency for IC_50_ values to be slightly lower for the kinase domain alone (Figure 8A), once K_i_ values were calculated, which incorporates the different ATP K_m_ values according to the equation
$$K_{i}=\frac{IC_{50}}{1+\frac{S}{K_{M}}}$$
no significant differences were seen between any of the constructs (Figure 8B).

## 3. Discussion

Biochemical studies of JAK proteins have traditionally been limited by the inability to express and purify significant amounts of recombinant material. We have overcome this limitation to characterize JAK1 enzymatically.

We found that the addition of the pseudokinase to the kinase domain significantly decreased k_cat_, (approximately 30-fold) confirming many in vivo studies that have shown that the pseudokinase domain has a negative influence on the catalytic activity of the kinase domain. The isolated kinase domain catalyzed phosphorylation at a rate of 0.4 events per second. This (relatively slow) rate is similar to that of many other kinases [42]; however, the full-length enzyme is significantly slower. Conversely, the presence of the pseudokinase domain decreased K_M_ (ATP), whereas K_M_ (peptide) remained unchanged. The most likely explanation for this is a direct pseudokinase-kinase domain contact, bringing about a conformational change in such a way that increases its affinity for ATP but not for STAT. This finding is consistent with the existing structure of the TYK2 pseudokinase-kinase domain construct [37]. The N lobe of the pseudokinase domain makes significant contacts with β1-β3 of the kinase domain N lobe, as well as the hinge region between the kinase domain N and C lobes (β5-αD). Molecular dynamics simulations on the serine/threonine kinase protein kinase A identified “communities” of residues involved in particular kinase functions [43]. β1–β3 of a kinase domain N lobe are in a community that is functionally important for ATP binding, consistent with our experimental findings. Increased affinity for ATP may also result in an increased affinity for ADP (the product of the phosphorylation reaction). As the rate-limiting step for many kinases is ADP release [42], this may be the reason behind the decreased catalytic activity of the full-length enzyme compared with the kinase domain alone. It is notable that the addition of the FERM/SH2 domains had no effect on any enzymatic parameters and that binding of a fragment of cytokine receptor likewise did not affect any catalytic constants. These observations suggest that there is no cross-talk between the N- and C-terminal halves of the full-length protein in vitro. However, in vivo JAK and the receptor are dimerized [44] and this may induce a conformation in which the FERM/SH2 can influence the activity if the catalytic domain.

Interestingly, we could not find a single enzymatic parameter that was affected by the presence of the V658F mutation. This included K_M_, k_cat_, activation rate, dephosphorylation rate, receptor binding affinity, or thermal stability. We conclude that, when studied in a model in vitro system, the mutation does not enhance the intrinsic activity of the kinase, the affinity with which it binds ATP, substrate or receptor, or induce any large-scale stabilization or destabilization. Despite this, in vivo, the mutation is clearly activating, as has previously been shown by many studies [27,45,46]. The prior body of work is more consistent with a model in which mutant JAK is hyper-activated (phosphorylated even in the absence of cytokine), rather than being hyperactive (having an increased catalytic rate, once activated). Our examination of the effects on K_M_ and k_cat_ (or lack thereof) supports the notion that the mutant enzyme, once activated, does not display an enhanced catalytic rate. Rather it is the activation process that goes awry, in vivo, in the presence of the mutation. Many cellular studies have shown the presence of a homodimeric receptor to be required for cytokine-independent activation of mutant JAKs [27,28]. In some cases, there is evidence that cytokine receptors exist as preformed dimers [47], and the orientation of JAKs with respect to those receptors is critical to the activation process [48]. In particular, the model favored by Brooks and Waters is that the pseudokinase exerts its inhibitory effect in trans [48]. Therefore, it is likely that the JAK1 V658F mutation only results in constitutive activation in the context of a fully assembled cytokine receptor complex, which was not present in our assays. Our data is consistent with a model in which the conformation of the receptor-kinase complex in cells is such that it prevents trans-activation (prior to cytokine exposure) and that it is this “brake” upon activation that is affected by the V658F mutation. Engineering a system in which intact cytokine receptors (correctly embedded in lipid bilayers) are present in auto-activation assays is technically challenging but our current efforts lie in this direction.

Whilst current generation JAK inhibitors have shown promise in the clinic and reduce the symptoms of the myeloproliferative disease, they have been limited by an inability to reduce the allelic burden of mutant forms of JAK, and by dose-limiting toxicities [49]. Although we found mutant JAK1 to be as susceptible to itacitinib (a small-molecule JAK1 inhibitor) as the wild-type enzyme, there was no evidence of it being any more susceptible. A JAK inhibitor which could selectively target mutant forms of the kinase could potentially overcome many of the limitations of current treatments. A greater structural and biochemical understanding of activating JAK mutations would assist with such a goal.

## 4. Materials and Methods

### 4.1. Cloning, Expression, and Purification of JAKs

N-terminally HIS-tagged human JAK1 constructs were cloned into the pFB_LIC_Bse plasmid using ligation independent cloning with the following amino acid domain boundaries: JAK1 kinase domain, 862-1154; FERM-SH2 domains, 25-559; pseudokinase-kinase domains, 566-1154; full length, 25-1154. Purified plasmids were used to make bacmid and transformed into baculovirus using standard techniques. *Spodoptera frugiperda* 21 (*Sf*21) cells were infected with baculovirus at a multiplicity of infection of 1. The cells were collected 48 h post-infection by centrifugation and frozen at −30 °C. Frozen cells from 1 L of cell culture were thawed and resuspended in 100 mL TBS/TCEP (10 mM Tris-HCl pH 7.5, 150 mM NaCl, 2 mM TCEP) supplemented with 1 mM phenyl-methylsulfonyl fluoride (PMSF) and 250 U DNAse. Cells were lysed by sonication and the lysate was cleared by centrifugation for 60 min at 50,000× *g*. Cleared lysate was filtered through a 0.8 μm filter and loaded onto a 1 mL HisTrap HP affinity column (GE Healthcare). The column was washed with Nickel Buffer A (20% (*v*/*v*) glycerol, 20 mM Tris- HCl pH 8.0, 500 mM NaCl, 5 mM imidazole, 2 mM TCEP) followed by 98% (*v*/*v*) Nickel Buffer A with 2%(*v*/*v*) Nickel Buffer B (20% (*v*/*v*) glycerol, 20 mM Tris-HCl pH 8.0, 500 mM NaCl, 500 mM imidazole, 2 mM TCEP), and then 93% (*v*/*v*) Nickel Buffer A with 7% (*v*/*v*) Nickel Buffer B. The protein was eluted from the column with 100% (*v*/*v*) Nickel Buffer B. Tags were cleaved overnight at 4 °C with TEV protease. Protein was loaded onto a Superdex 200 16/600 size exclusion column in JAK Gel Filtration Buffer (10% (*v*/*v*) glycerol, 20 mM Tris-HCl pH 8.0, 500 mM NaCl, 2 mM TCEP). Fractions were assayed by reducing SDS-PAGE gel and those containing pure complex (>70%) were pooled and concentrated to ~2 mg/mL. For activated JAKs, ATP, and MgCl_2_ were added to 1 mM and 2 mM respectively and protein was incubated for 24 h at 8 °C.

### 4.2. Kinase Assay

The 10 µL reaction mix contained 0.5 µL 20× Kinase Assay Buffer, 1 µCi γ-^32^P-ATP and varying concentrations of ATP, STAT5b peptide, and inhibitor, as described for each experiment. Reactions were started with the addition of 5 µL JAK solution, with concentration as described for each experiment. After incubation at room temperature for 20–60 min, a 3.5 µL reaction mix was spotted onto P81 phosphocellulose paper (Dr. J. Oakhill, SVIMR. Australia) and the paper was placed into 5% (*v*/*v*) H_3_PO_4_. Paper was washed 4 × 15 min with 100 mL 5% (*v*/*v*) H_3_PO_4_, followed by 1 × 1 min with acetone. Paper was dried and exposed to a phosphorimager plate overnight. The plate was scanned using a Typhoon FLA 7000 phosphorimager (GE Life Sciences, Chicago, IL, USA) and the incorporated radioactivity of each spot was quantitated using ImageQuant TL software (GE Life Sciences). The background radioactivity, quantified from an area of the paper without reaction mix, was subtracted from that for the experimental spots.

### 4.3. Activation Assay

JAK1 was diluted to 1.1× the indicated concentration in the JAK GF buffer. 10% (*v*/*v*) 10 mM ATP, 20 mM MgCl_2_ was added to begin the reaction. At time points, an aliquot of reaction mix was added to 4× SDS-PAGE reducing buffer (50 mM Tris-HCl pH 7.4, 200 mM β-ME, 10% (*v*/*v*) glycerol, 4% (*v*/*v*) SDS, 0.2% (*w*/*v*) bromophenol blue) to yield a final protein concentration of 300 nM. The sample was run on an SDS-PAGE gel and transferred to PVDF membrane. Western blot was performed using an anti JAK1 pY1034/1035 antibody (Santa Cruz Biotech sc-10176, 1/1000, Dallas, TX, USA) and an IR fluorescent secondary antibody (LI-COR IRDye 800 CW 925-32211, 1/15,000). Band intensities were quantified using an Odyssey IR imaging system (LI-COR).

### 4.4. Deactivation Assay

Phosphorylated JAK1 was mixed with 900 nM purified recombinant PTP1B and 30 mM EDTA in the JAK GF buffer. Reaction timepoints were taken and assayed by western blot as described above for activation assays.

### 4.5. Thermal Shift Assay

JAKs of 3–5 µg were diluted into the JAK Gel Filtration Buffer with 0.04× SYPRO orange, 4% (*v*/*v*) DMSO plus either 20 mM EDTA, 1 mM ATP and 2 mM MgCl_2_, or 6 µM itacitinib, and with or without 5 µM receptor peptide. The temperature was raised in 1 °C per min steps from 25 °C to 80 °C and fluorescence readings at 530 nm were taken at each interval. Sigmoidal curves were fitted to the fluorescence data to provide T_m_ values for each experiment.

### 4.6. Data Fitting

Active enzyme concentrations [*E*] were fitted to the Morrison equation:$$\frac{V_{i}}{V_{0}}=1-\frac{\left(\left[E\right]+\left[I\right]+K_{i}^{app}\right)-\sqrt{{\left(\left[E\right]+\left[I\right]+K_{i}^{app}\right)}^{2}+4\left[E\right]\left[I\right]}}{2\left[E\right]}$$
where
$$K_{i}^{app}=K_{i}\left(1+\frac{\left[S\right]}{K_{M}}\right)$$
and [*E*] is JAK1 concentration, [*I*] is the itacitinib concentration, [*S*] is the ATP concentration, and *K_M_* refers to the ATP *K_M_* of JAK1.

STAT *K_M_* data were fitted to the Michaelis–Menten equation:$$V=\left(\frac{V_{\max\;}\left[S\right]}{\left[S\right]+K_{M}}\right)$$
where [*S*] represents the concentration of the substrate whose K_M_ was being measured.

*k_cat_* data were fitted to the equation:$$k_{cat}=\frac{V_{max}}{E_{t}}$$
where *E_t_* represents the enzyme concentration used in the reaction as determined by Morrison kinetics.

Itacitinib *K_i_* data were fitted to the equation:$$K_{i}=\frac{IC_{50}}{1+\frac{S}{K_{M}}}$$
where [*S*] represents the ATP concentration and *K_M_* the *K_M_* for ATP.

## 5. Conclusions

In conclusion, our findings are that the mutation in JAK that causes myeloproliferative disease does not alter the enzymatic characteristics of the kinase once it is activated. Rather, it seems likely that a failure of the processes keeping the enzyme “switched-off” in the absence of cytokine (and hence leading to constitutive activation) is the basis of the molecular phenotype induced by the mutation. These results suggest that standard kinase-inhibitors which bind the ATP binding site could likely not target the mutant enzyme selectively. This highlights the need for detailed structural information on full-length JAK in its wild-type and mutant forms. 

## Figures and Tables

**Figure 1 cancers-11-01701-f001:**
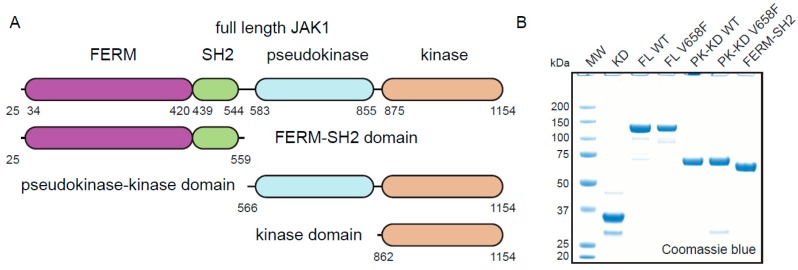
Purification of JAK1 constructs. (**A**), Schematic representation of the JAK1 domains used for expression. The internal domain boundaries are numbered on the full-length construct, whilst N and C terminal boundaries are numbered at each end. (**B**), Purified JAK1 constructs shown on SDS-PAGE gel stained with Coomassie blue. KD = kinase domain, FL WT = full length wild type JAK1, FL V658F = full length JAK1^V658F^, PK-KD WT = pseudokinase-kinase domain wild type, PK-KD V658F = pseudokinase-kinase domain V658F, FERM-SH2 = FERM-SH2 domain.

**Figure 2 cancers-11-01701-f002:**
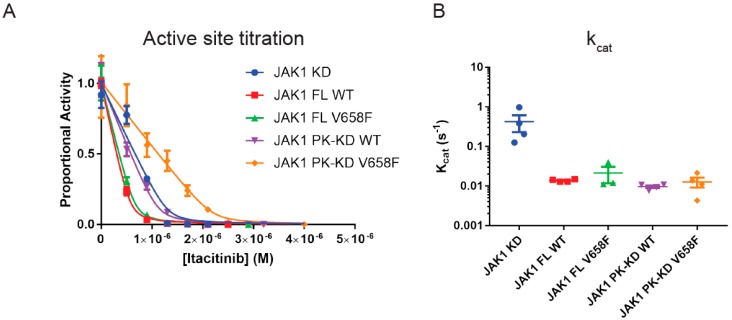
The pseudokinase domain reduces the k_cat_ of the catalytic domain. (**A**), Representative Morrison active site titration of JAK1 constructs to quantify levels of active enzyme. Error bars represent the range of two measurements. (**B**), summary quantifications of JAK1 phosphorylation of STAT peptide substrate at 1 mM ATP, 5 mM STAT, controlled for reaction time and active enzyme concentration. The addition of the pseudokinase domain reduced kinase activity compared to the kinase domain alone. Error bars represent standard error of the mean of four experiments (*n* = 4). Construct abbreviations are as per Figure 1.

**Figure 3 cancers-11-01701-f003:**
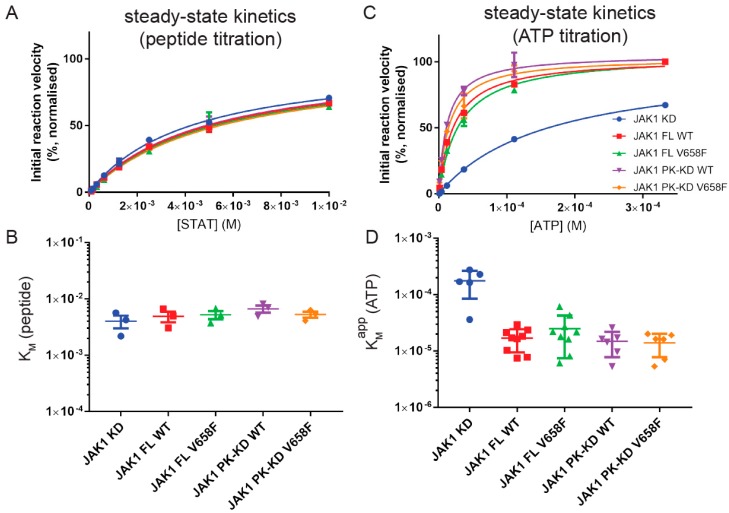
The V658F mutation does not affect ATP or peptide K_M_. (**A**), Representative kinase activity assay with STAT titration at 0.5 mM ATP. Error bars represent the range of two measurements. (**B**), Summary quantification of K_M_ values calculated from independent STAT titrations. No significant difference in STAT K_M_ was observed between JAK1 constructs. Error bars represent the standard error of the mean from three independent experiments (*n* = 3). (**C**), Representative kinase activity assay with ATP titration at 6 mM STAT. Error bars represent the range of two measurements. (**D**), Summary quantification of K_M_ values calculated from independent STAT titrations. JAK1 constructs containing the pseudokinase domain exhibited a lower ATP K_M_ than the kinase domain alone. Error bars represent the standard error of the mean from five to nine independent experiments. Construct abbreviations are as per Figure 1.

**Figure 4 cancers-11-01701-f004:**
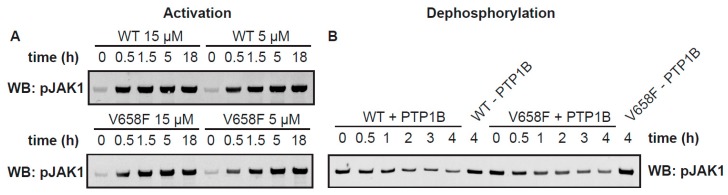
The V658F mutation does not affect phosphorylation and dephosphorylation rates. (**A**), In vitro phosphorylation of purified full-length JAK1 wild type (WT) or V658F over time at 15 µM and 5 µM JAK1, as determined by western blot for JAK1 pY1034/1035. Activation loop phosphorylation occurs faster at 15 µM JAK1 than 5 µM; however, it is unaffected by the presence of the V658F mutation. (**B**), In vitro dephosphorylation of purified full-length JAK1 wild type (WT) or V658F with purified phosphatase PTP1B, as determined by western blot for JAK1 pY1034/1035. Dephosphorylation of JAK1 over time is unaffected by the presence of the V658F mutation. Detailed information can be found at Appendix A.

**Figure 5 cancers-11-01701-f005:**
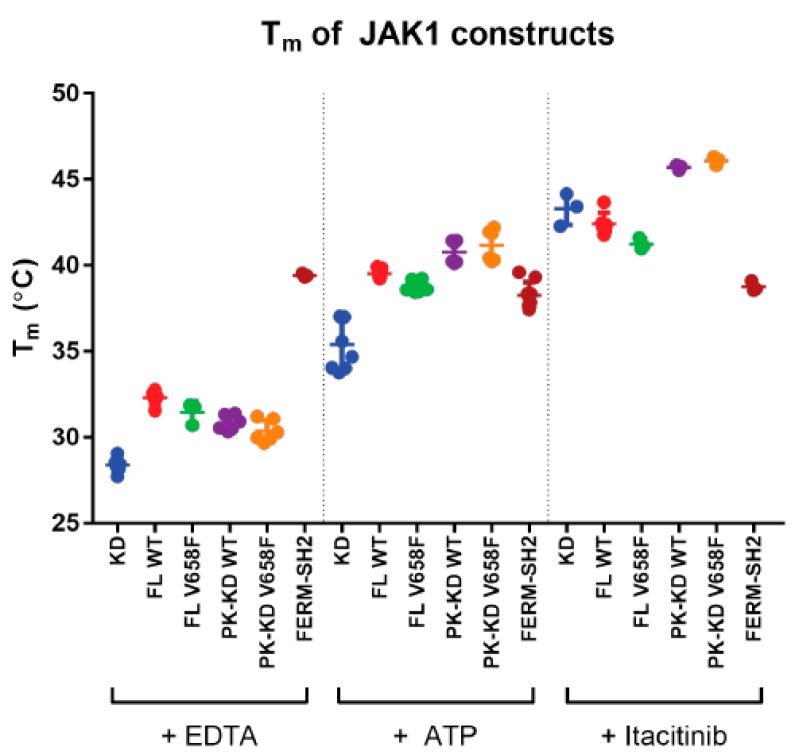
The V658F mutation does not affect the thermal stability of JAK1. Summary of melting temperatures of JAK1 constructs with an empty ATP binding site (+ EDTA), with ATP bound (+ ATP), or with a small molecule JAK inhibitor bound (+ Itacitinib). All kinase and pseudokinase domain-containing constructs show an increased T_m_ upon the addition of ATP or itacitinib. There were no significant differences in T_m_ between equivalent constructs with or without the V658F mutation. Error bars represent standard error of the mean of two to five independent experiments (*n* = 2–5). Construct abbreviations are as per Figure 1.

**Figure 6 cancers-11-01701-f006:**
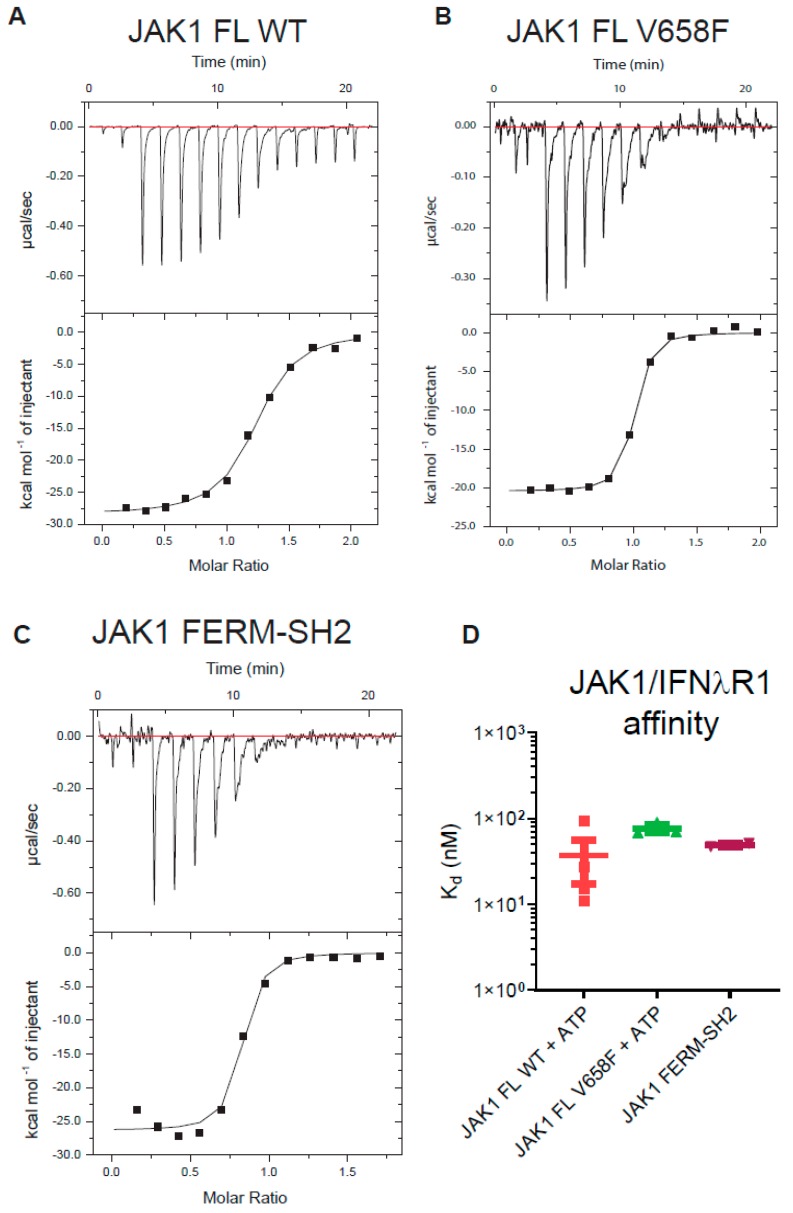
The V658F mutation does not affect JAK1 binding to the box1/2 motif of a cytokine receptor. (**A–C**), Representative ITC curves of IFNλR peptide binding to JAK1 constructs (**A**) full-length JAK1; (**B**) full-length JAK1^V658F^; (**C**) JAK1 FERM-SH2 domains only. (**D**), Summary of IFNλR-JAK1 K_d_ values obtained from ITC experiments. No significant difference in IFNλR affinity was seen between full-length wild type, full-length V658F, nor the FERM-SH2 domains of JAK1. Error bars represent the standard error of the mean of three independent experiments (*n* = 3).

**Figure 7 cancers-11-01701-f007:**
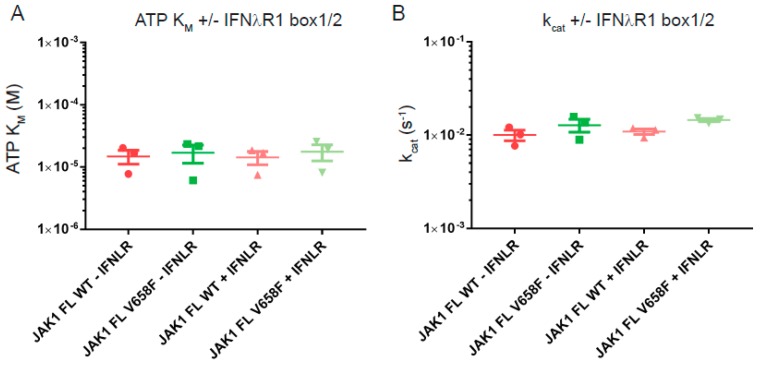
Binding to the interferon lambda box1-box2 peptide does not affect JAK1 or JAK1 V658F kinetic parameters. (**A**), Summary of ATP K_M_ values from kinase activity assays in the presence and absence of 2 µM IFNλR peptide. IFNλR binding did not result in a significantly different ATP K_M_ for JAK1. (**B**), Summary of k_cat_ values from kinase activity assays in the presence and absence of 2 µM IFNλR peptide. IFNλR binding did not result in a significantly different ATP k_cat_ for JAK1. Error bars represent standard error of the mean from three independent experiments (*n* = 3).

**Figure 8 cancers-11-01701-f008:**
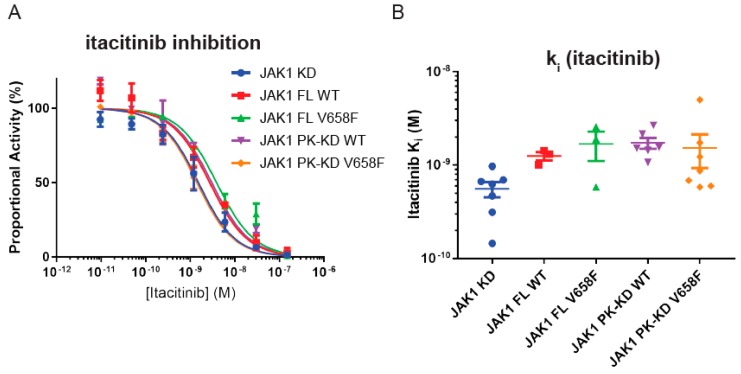
The V658F mutation does not affect inhibition by a small-molecule JAK1 inhibitor. (**A**), Representative IC_50_ curve from in vitro kinase assays with a titration of Itacitinib. Error bars represent the range of two measurements. (**B**), Summary quantification of k_i_ data after correction for differing ATP K_M_ values between different JAK1 constructs. No significant differences in K_i_ were observed between constructs. Error bars represent the standard error of the mean of three to seven independent experiments.

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
