# Peer review of "Enzymatic Characterization of Wild-Type and Mutant Janus Kinase 1"

_cancers, 2019, doi:10.3390/cancers11111701_

Round 1
Reviewer 1 Report
1) Abstract: The first word is mistakenly typed as nus instead of Janus
2) What is the difference between the members of the JAK family? Which is the predominant JAK in the body based on expression and/or function? How conserved are the sequences among the 4 JAK family members?
3) What is the frequency of occurrence of mutations in the JAKs? What other mutations are prevelant in this family of enzymes?
4) What assays were done to validate that the protein isolated is JAK1? What do the authors think they did differently to be successful in isolating full-length JAK1 and on not being to replicate it with JAK2?
5) Did the authors test any other mutations?
6) What functional assays do the authors plan to do as the next steps? If there are specific domain inhibitors available then the enzymatic/functional assays need to be carried out to identify what effect does the mutations really have?
Reviewer 2 Report
Manuscript “Enzymatic characterization of wild-type and mutant Janus Kinase 1” describes purification and enzymatic characterization of full length JAK1 wild type and mutant JAK1V658F. Mutations in pseudokinase domain of JAKs are underlying cause of human myeloproliferative diseases. Detailed biochemical characterization of mutant JAKs forms opens possibility for more targeted design of therapeutics and thus provides valuable information. Moreover I would like to stress that authors, for the first time, managed to purify full length active JAK1, what underlines the value of the presented study. Given all the above I recommend the manuscript for publication in Cancers. The study is properly designed, performed and described, but there are several minor amendments that can improve scientific value of the manuscript.
Specific comments related to the manuscript:
In my opinion Western blot of Figure 4A in fact shows that mutant V658F shows slower activation than the wild type form – at 0.5 h band pJAK1 band is weaker for V658F than for the WT. It will be helpful if the authors estimate intensities of the bands and provide them as bar graph. It seems that 0.5 h time point is too long as for the WT signal already reaches saturation (compare band intensities at time points 1.5 and 1.5 h). Maybe better kinetics of the phosphorylation increase would be observed if shorter time points will be tested. On Figure 8A description of samples should be corrected to match the description on other figures. Including functional assays on cell lines that would give more information on functional consequences V658F mutation would improve scientific value of the manusript.
